



# On the Response of the Equatorial Atmosphere and Ocean to Changes in Sea Surface Temperature along the Path of the North Equatorial Counter Current

David J. Webb[1]

[1]National Oceanography Centre, Southampton SO14 3ZH, U.K.

**Correspondence:** David J. Webb (djw@noc.ac.uk)

**Abstract.** The CESM climate model is used to test the hypothesis that the changes observed during El Niños are, at least in part, a response of the coupled ocean/atmosphere system to changes in sea surface temperature along the path of the North Equatorial Counter Current.

The model results show that increased temperatures at the latitudes of the NECC produce more deep atmospheric convection
within the ITCZ. This has a local effect on the ocean's surface pressure field. The increased deep atmospheric convection also appears to affect the longitude structure of the Hadley Circulation. This results in less sinking air in the south-east Pacific and produces changes in surface pressure similar to those of a Southern Oscillation. Both mechanisms reduce the zonal component of the surface pressure gradient and wind stress along the Equator, and produce an El Niño type response in the ocean.

## 1 Introduction

During a period when studies of El Niños generally concentrated on the seas close to South America, Wyrtki (1973, 1974) used sea level data from Pacific Islands to show that, in the western Pacific, the transport of the North Equatorial Counter Current (NECC) increased during El Niño events. His studies led to a wider interest in the subject and eventually to our understanding that El Niños are responsible for events both locally in the Pacific and at global scales (McPhaden et al., 2020).

Since that early there has been a huge amount of research on the El Niño and related problems, many of which are also
discussed in McPhaden et al. (2020). In the atmosphere, much of the research focusses on the generation and propagation of Rossby waves, which interact with the jet streams and affect weather patterns over large parts of the globe (Trenberth et al., 1997).

The Rossby waves may be generated by the high level divergences generated by deep atmospheric convection near the Equator (Sardeshmukh and Hoskins, 1988). However it is difficult to generate such waves in the equatorial easterly winds.
Thus Rasmussen and Mo (1993) found only weak Rossby wave sources within 15 degrees of the Equator. In contrast they found that the strongest Rossby wave sources were associated with the convergence regions associated with the descending branch of the Hadley Cell.

In the ocean, the focus has been on the equatorial band and changes in the structure of the thermocline and equatorial currents. In a normal year easterly winds over the Equator generate an east to west slope in sea level and a similar increase in





depth of both the surface thermocline and the depth of the Equatorial Undercurrent. The winds also cause upwelling along the
Equator, generating the eastern Pacific cold pool.

During El Niño events, the easterly wind weakens, there is a relaxation of thermocline slope, a change in the undercurrent,
and reduced upwelling on the Equator, leading to a warmer cold pool.

West of the dateline, the easterlies on the Equator are replaced by westerlies. The resulting Ekman transport then converges
warm pool water onto the Equator where the wind now generates an eastward flowing Equatorial Current. This carries the
warm water towards the dateline where a region of deep atmospheric convection develops.

A number of mechanisms have been proposed as the cause of El Niños. A common assumption is that the El Niño-La Nino
cycle is an unstable mode of the coupled ocean-atmosphere system. In the ocean this is often described interms of a discharge-
recharge mode, the warm pool-of the west Pacific being discharged during each El Niño event and then being recharged in the
years preceding the next event.

Early theories proposed that the events were triggered by equatorial Kelvin and Rossby waves in the ocean but at present the
two main contenders are Madden-Julian oscillations and westerly wind bursts.

Both have been observed in the western Pacific prior to El Niños and both could move warm warm pool water towards
the dateline. But as stated by Yu and Ferorov (2022), when discussing westerly wind bursts, "quantifying their effects is
challenging".

Published research on El Niños usually concentrates on the Equatorial band and there have been few papers on the NECC
and the nearby Inter-Tropical Convection Zone (ITCZ). However as well as Wyrtki's early papers, Rasmussen and Carpenter
(1982) reported changes in the winds associated with the ITCZ and its southward movement during an El Niño.

Others also studied changes to the NECC and the ITCZ during El Niño events but, although correlations can only provide a
statistical connection, not a causal one, the discussion is often in terms of the passive response of the NECC and the ITCZ to
the El Niño signal (i.e. Zhao et al. (2013) and Zhou et al. (2021)), not their possible active role.

## 1.1  The Present Paper

The present paper is a development from Webb (2018), a study of the equatorial currents during the strong El Niños of 1982-83,
1997-98 and 2015-16. The study used data from a high resolution ocean model and found that the NECC was much warmer
than normal during the development of strong El Niños.

The study also reported that the warm water came from the warm pool in the western Pacific. The water was carried eastwards
by a stronger than normal NECC, starting in the west during the northern spring and arriving in the eastern Pacific late in the
year.

Although the discharge mechanism is often decribed in terms of gravity driven currents along the Equator, this study indi-
cated that in the cases studied, much of the west Pacific warm pool is best described as being 'extracted' by the geostrophically
blanced North Equatorial Counter Current.





A follow up study using satellite data (Webb et al., 2020) confirmed the findings, and a second modelling study (Webb, 2021) showed that it was the winds in the western Pacific early in the year which were responsible for the initial increase of transport by the NECC.

This connection between the winds in the western Pacific early in the year and the development of a strong El Niño late in the year could be explained if it is the warm NECC itself which is at least partly responsible for the strong El Niño. The time required to advect water of across the Pacific would also help explain the long timescales involved in the fluctuations of El Niño indices.

## 1.2 Hypothesis

Based on the comments above, this paper concentrates on testing the hypothesis that strong El Niños are due, at least in part, to warmer than normal water along the path of the NECC.

The study also tests a hypothesis that the close relationship between the NECC and ITCZ is involved. The NECC lies just south of the latitude of the ITCZ. Convection in the ITCZ draws in surface air from the north and south, but that from the south crosses the NECC and so is readily affected by any changes in sea surface temperature.

Temperatures in the NECC are usually above 26°C, close to the temperature range in which deep atmospheric convection is observed over the ocean (Gadgil et al., 1984; Evans and Webster, 2014; Kubar and Jiang, 2019). In addition the remoteness from other centres of deep convection, may make the region more sensitive than normal to changes in sea surface temperature (SST) (Williams et al., 2023).

The energy available to drive deep convection depends primarily on atmospheric water content. Over the ocean relative 75  humidity is high, around 80%, so the total water content is, to first order, a function of temperature, a one degree rise increasing the water content by 7% or more (Biri et al., 2023). As a result, even small increases in NECC temperature can have a significant effect on convection in the nearby ITCZ.

Increased convection in the ITCZ will lower near-surface atmospheric pressure but the size of the region affected is not immediately obvious.

The effect of convective cells at and close to the Equator was first discussed by Matsuno (1976) and later Gill (1980), using an analytic model which represented the first vertical mode of the atmosphere. They also only considered the case of convective cells with small horizontal scale, whereas the ITCZ extends for thousands of kilometers.

A convective source, closer in structure to that of the ITCZ, is discussed by Vallis (2017). He provides the first mode solution for an infinitely long line of convection at a fixed latitude. With no background wind field, this shows that there is a band of 85  westerly winds to the south of the line of convection. The width of the band depends on the Equatorial Rossby radius $L_{eq}$, and the time scale for development depends on $T_{eq}$ where.

$$L_{eq} = (c/(2\beta))^{1/2}. \tag{1}$$
$$T_{eq} = (2c\beta)^{1/2}. \tag{2}$$





Here $c$ is the speed of the first (and fastest) vertical mode of the atmosphere and $\beta$ is the gradient of the Coriolis term with
latitude. If $c$ is 25 ms$^{-1}$ then $L_{eq}$ corresponds to a distance of around 7 degrees of latitude and $T_{eq}$ corresponds to about 8
hours. If the estimates are correct then increases in ITCZ convection a few degrees north of the Equator will affect winds on
the Equator within a few hours.

Given the strong easterly winds normally found on the Equator, the changed pressure field may not be sufficient to change
the direction of the winds, but even a small reduction in wind stress would affect the east-west surface slope of the ocean and
so may produce other El Niño type changes.

As a thought experiment with simple quasi-linear logic, this seems reasonable. But the climate system can be highly non-
linear, so the hypothesis needs testing in a more realistic environment.

Here, this is done by setting up a response experiment using the NCAR Community Earth System Model (CESM). A simple
forcing is used in which the temperature along part of the NECC is increased slightly. The paper then reports on the resulting
changes in ITCZ convection and the changes in atmospheric pressure, equatorial winds and the equatorial ocean, following the
chain of events proposed above.

The test does not check that the NECC is responsible for the timescales of El Niños, but it does check that a typical NECC
temperature fluctuation is sufficient to produce a recognisable El Niño signal.

In the rest of the paper, section 2 gives details of the CESM model and the forcing condition used. Sections 3 and 4 then
report on the response of the atmosphere and ocean. Finally section 5 discusses the wider implications of the study.

## 2   Model and Forcing Experiment

This study makes use of version 2.1.3 of CESM, the NCAR Community Earth System Model (Hurrell et al., 2013; Lauritzen
et al., 2018). The model is widely used by the earth science community for global change research (Schneider et al., 2022;
Holland et al., 2024), and for studies of individual parts of the climate system (Li et al., 2018; Dolores-Tesillos et al., 2022;
Maher et al., 2023).

The model code is supported by a large number of component sets which are useful for different climate studies. The
component set used for the present study is 'B1850', a basic set in which all but one the sub-models (atmosphere, ocean, ice,
land) are active. The outlier is the wave model in which the ocean surface wave field is fixed.

For the present study, the two key components are the ocean and the atmosphere. The atmosphere has 32 levels in the
vertical, and resolutions of 1.25 degrees in longitude and 0.934 degrees in latitude. The ocean has 60 layers in the vertical and
a longitude resolution of 1.1 degrees. Within 20 degrees of the Equator the latitude resolution 0.26 degrees, extending to 1.125
degrees nearer the poles. The level thickness are 10 m in the top 200 m of ocean, but then increasing with depth.

The ocean model resolution means that it should be able to represent most of the small scale features of the tropical ocean,
but the longitude resolution may be too large for a good representation of tropical instability waves. The model produces a
reasonable El Niño response (Capotondi et al., 2020) but, like many other similar models, its NINO 3.4 index does not show
the asymmetry between El Niño and La Nina states, seen in reality.



Many couple climate models suffer from the double ITCZ problem (Song and Zhang, 2019) making comparison against observations difficult. In the present relatively short runs of CESM model version 2.1.3, there is no evidence of a double ITCZ.

## 2.1 Forcing Experiment

The present study is based on a control run and a forced run. In the control run, the model was started with the default B1850 initial fields and parameters, and run for a period of 10 years. At the end of each month, the model archived the previous month's averages of key variables. It also generated a set of restart files for the following month.

During the first eighteen months of the control run the Niño 3.4 index was large, around 2 degrees, but by year 5 this had dropped to values near zero. Previous studies (Webb, 2018; Webb et al., 2020; Webb, 2021) showed that during the development

of strong El Niños, the period around September appeared to be an important growth period, with NECC temperatures around 30°C between 180°E and 240°E.

The forced run is initialised using the control run restart data for the 1st August of year 5. Near surface temperatures in the region between 180°E and 240°E and between 4°N and 7.5°N, are then relaxed towards values which match the normal zonal gradient of SST. At 180°, the forcing is towards 29.5°C. At 240°, it is towards 28.5°C, with a linear dependence in between.

These temperatures are slightly above those of the control run at the beginning of August.

In the top 30 m of ocean, the forced run uses a decay time of 2 days. Below this, the relaxation coefficient, the inverse of the decay time, decreases linearly with depth to a value of zero at 100 m.

The forcing region is slightly to the south of the path of the NECC in the control run, but corresponds to the latitude of the NECC during a strong El Niño (Webb, 2018). During August the average surface temperature within the forcing region is

28.81°C in the forced run, and 27.75°C in the control run. During September the values are 28.91°C and 27.54°C.

## 3 Response of the Atmosphere

The results, presented in the next two sections, focus on the average response of the atmosphere and ocean during September of year five. The month is chosen because, by the centre of the month, six weeks after the start, the main atmospheric response should have setted down and there has been sufficient time for the ocean response to develop. Also seasonal changes should

not be a major factor in the response.

### 3.1 Atmospheric Convection

Figure 1 shows the the rate of change of atmospheric pressure following particles crossing the 525 hPa surface in the control and forced runs. Negative values correspond to regions of convection and positive values to regions of sinking.

The ITCZ shows as a band of enhanced convection north of the Equator, bounded roughly by longitudes 150°E and 270°E.

In the east it merges with the area of convection lying above the the East Pacific Warm Pool.



The result from the control run is in reasonable agreement with results from the ECMWF ERA5 reanalysis[1]. Convection over the ocean to the east of New Guinea is weaker than expected and the gap in the ITCZ around 240°E may also be unusual.

The area in the central Pacific, where the temperature is modified during the forced run, is shown by the rectangle lying north of the Equator. In this region the results show that, for the forced run, convection in the middle atmosphere increases slightly
and the ITCZ band moves slightly south. Near 240°E the linkage to the East Pacific Warm Pool is strengthened, but west of 150° convection becomes weaker.

A summary of the fluxes in the two runs and their variation with height is given in Table 1. If $F$ is the integrated mass flux through a pressure surface, then

$$F \quad = \quad -(1/g) \; \int (dP/dt) \, R^2 \, sin(\phi) \, d\psi \, d\phi, \tag{3}$$

where $dP/dt$ is the Lagrangian flux through a pressure surface, $\psi$ is longitude, $\phi$ latitude, $g$ gravity and $R$ the radius of the Earth.

The latitude integral is between the Equator and 20°N, the range being chosen to include the whole of the rising branch of the Hadley Cell during the late northern summer. The integral over longitude is either for the full 360 degrees, or for just the forcing region between 180°E and 240°E.
A second integral, $F_{W>0}$, includes only the convecting regions,

$$F_{W>0} \quad = \quad -(1/g) \; \int (dP/dt) \, \theta(-dP/dt) \, R^2 \, sin(\phi) \, d\psi \, d\phi, \tag{4}$$

where,

$$\theta(x) \quad = \quad 1 \quad if \; x > 0,$$
$$\quad = \quad 0 \quad otherwise$$

In the control run, the total vertical flux varies only slightly between 821 hPa and 322 hPa whichever flux integral is used. However in the integration over the forcing longitudes, the fluxes drop off rapidly with height, the flux at 322 hPa being 20% of that at 821 hPa in the full integral and less than half when only integrating over convective regions.

These results are consistent with ERA5 reanalysis data, Webb (2023) finding that during an average year ITCZ detrainment in the middle atmosphere is roughly balanced by entrainment over the Island Continent and that this limits the reduction of
convective flux with height.

In the forced run, the increased sea surface temperature might be expected to increase convection. However at the lowest level, 821 hPa, all the integrals show that convection in the forced run is slightly reduced. The reason for this is not known. A similar slight reduction is also found higher in the atmosphere, at 525 hPa and 322 hPa, in the integrals over all longitudes.

However in the forced region, between 140°E and 240°EA, the behaviour is different, with the flux in the forced run dropping
off much more slowly with height than in the control run. As a result at 500 hPa and above, the forcing region becomes a much greater contributor to the total vertical flux than in the control run.

---

[1]Webb (2023) includes figures and flux estimates similar to the ones presented here but based on ERA5.





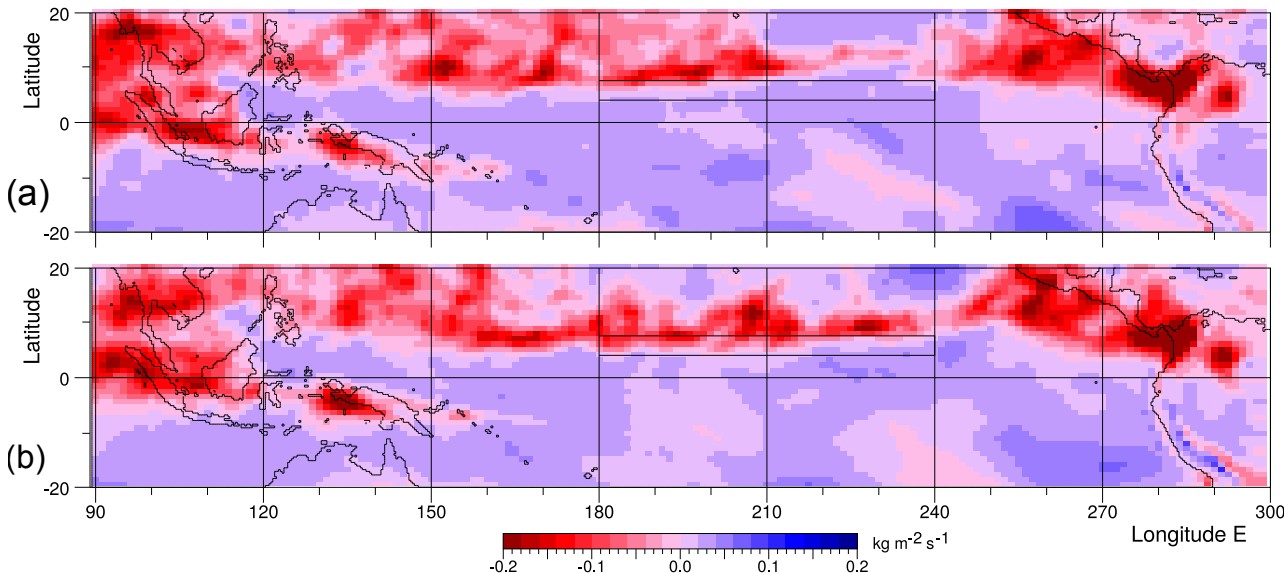

**Figure 1.** Average of the Lagrangian flux ($dP/dt$) across the 525 hPa pressure surface in September of year 5 for (a) the control run and (b) the forced run. The forcing, described in the text, starts at the beginning of August in year 5 and is non-zero within the marked rectangular region in the central Pacific. Negative (red) values correspond to convection.

**Table 1.** Average integrated vertical mass fluxes ($\text{Tg s}^{-1}$) in the control and forced runs during September of year 5. The integrals are between the Equator and 20°N. Different columns correspond to either the whole globe or the forcing longitudes (180°E to 240°E) and are for either the full integral or for the integral only over convecting regions (i.e. where $dP/dt < 0$, $P$ being the pressure following a particle).

| Level hPa | Run | Zonal Average Total | $dP/dt < 0$ | 180°E-240°E Total | $dP/dt < 0$ |
|---|---|---|---|---|---|
| 322 | Control | 248.5 | 330.5 | 11.9 | 32.8 |
|  | Forced | 240.4 | 320.4 | 26.6 | 48.9 |
| 525 | Control | 260.1 | 334.2 | 16.5 | 38.1 |
|  | Forced | 255.8 | 327.9 | 32.1 | 48.9 |
| 821 | Control | 254.7 | 330.4 | 51.4 | 73.3 |
|  | Forced | 250.1 | 320.3 | 50.0 | 69.7 |

This again has some similarity with the ERA5 data, Webb (2023) finding that during strong El Niños, when the temperature of the NECC is high, deep atmospheric convection over the ITCZ increases and entrainment over the Island Continent decreases.





### 3.1.1 ITCZ Convection and the Vertical Modes

Although the reduction in convection during the forced run, discussed earlier, was unexpected, the fact that increased sea surface temperatures results in increased deep atmospheric convection could be important.

This is because Salby and Garcia (1987) found that convection near the Equator excites modes with a vertical scale about twice the height of the convective region. De-Leon et al. (2020) also reported that tropical waves, forced by convective maxima in the mid-troposphere, project strongly onto low vertical modes, the ones with the fastest phase speeds and the largest horizontal scales.

As a result of the greater amount of deep atmospheric convection in the forced run, it should trigger more of the lowest order atmospheric modes and so have a greater horizontal impact.

As the scale length of the lowest mode is expected to be around 6 degrees, and the ITCZ lies a similar distance from the Equator, it is possible that sea level pressures at the Equator are affected by the low lea level pressure along the line of the ITCZ. This possibility is considered next.

### 3.2 Atmospheric Pressure

Figure 2 shows the average surface pressure in the control and forced runs, together with the difference. Near the Equator zonal pressure gradients are very weak, a difference of around 4 hPa over 180 degrees of longitude (~20,000 km). Meridionally they are also small, with a difference of around 2 hPa between the Equator and the ITCZ trough at 10°N.

In the forced run, pressures along the line of the ITCZ are reduced, but the average change is less than 1 hPa. Within the forcing region there is an narrow valley of reduced pressures. This is orientated in an east-west direction and has a north-south extent comparable to the width of the forcing region. The depth of the valley is of order 0.02 hPa. Thus it appears to be a short range local response to the changes in convection.

Around this valley lies a broader minimum extending to 10°N. In the south its extent is more variable, extending to the Equator at 180°E and 240°E, and to 10°S near 210°S. In terms of the earlier analysis, this probably results from the increased penetration depth of convection along the line of the ITCZ.

However, as best seen in the difference figure, the major contribution to the surface pressure field come from other processes. In the east, the reduction in surface pressure on the Equator is connected to a much larger drop in surface pressures in the southeast Pacific. In the west, the rise in surface pressure on the Equator is connected to mush larger increases to the north and south near the longitudes of the dateline.

It is tempting to connect the latter maxima with the Rossby wave part of the solution described by Gill (1980). If they are a Rossby wave response to the forcing, a larger forcing should result in easterly winds in the western equatorial Pacific, as is observed during El Niño events.

However, in the south, the centre of the pressure maximum is near 25°, corresponding to an equatorial scale of over 12 degrees or a wave speed of over $80 \text{ ms}^{-1}$. This is unlikely. In the north there, is no similar maximum until 40°N. Thus although both features may be related to Rossby waves, for the moment the result is not conclusive.




**Figure 2.** Averaged sea level pressure in September of year 5 for (a) the control run (b) the forced run, both with contour interval of 0.5 hPa, and (c) the change due to the forcing, with contour interval of 0.25 hPa.

There is also a problem in the south-east Pacific where the minimum in the pressure is not predicted by theories of either equatorial deep convection (Matsuno, 1976; Gill, 1980; Vallis, 2017) or the Hadley Cell (Held and Hou, 1980; Schneider, 2006; Hoskins et al., 2020). This is discussed next.

### 3.3 The Large-Scale Response

Figure 3 is a global version of the figure showing the change in atmospheric pressure due to the forcing. Although the forcing is only of order 1°C and is over a limited area, it has generated pressure changes of more than 5 hPa over large areas of ocean. In the South Pacific, the mean pressure difference between 30°S and the Equator is generally less than 15 hPa, so the change is large.





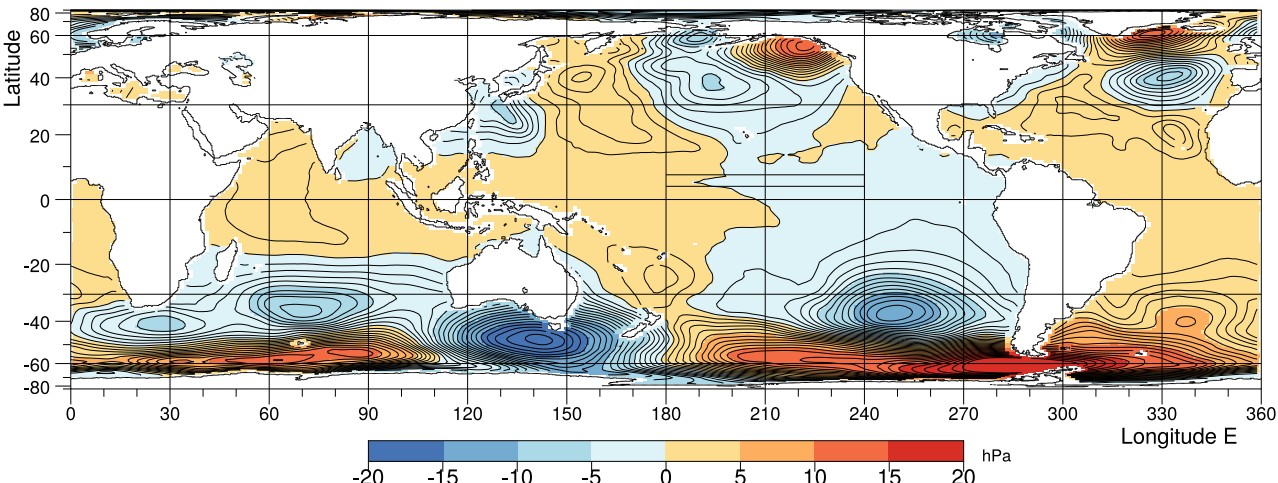

**Figure 3.** Global change in sea level pressure, due to the forcing, averaged over September of year 5. Contour interval 1 hPa.

During the northern summer, the high pressure region in the south-east Pacific is one of the main Hadley Cell sinking regions. As summer comes to an end, pressures in the region might normally drop, but here the forced run is showing an extra reduction, which continues in the following months.

In retrospect it is possible that such a strong response should have been expected. The earlier studies (Webb, 2018, 2023) showed that El Niños were associated with both a warm NECC, a reduction in deep atmospheric convection over the Island Continent and an increase over the ITCZ.

As both regions are important contributors to the Hadley Cell, it seems inevitable that a change in the convection longitudes, as seen in the forced run, will affect the strength of the descending branches of the Cell. Global plots of the vertical flux in the atmosphere show that sinking in the south-east Pacific is reduced in the forced run.

This reduction then helps to explain the surprising result that increased deep convection along the line of the ITCZ results in reduced surface pressures in the South Pacific. As discussed in the next section this has a significant effect on the ocean surface winds.

However before leaving the global scales, it is noticeable that the changes in model surface pressure in the Pacific region are very similar to the Trenberth and Caron (2000) Southern Oscillation correlation patterns during the autumn months. Both show a maximum in the south-east Pacific extending to the Equator and a minimum over the Island Continent. Both also show a second maximum in the North Pacific.

In summary, the model results imply that the increased temperature along the line of the NECC affects the pressures close to the Equator in three ways. The first is a direct path, the influence of surface pressure changes below the ITCZ. The second is via increased surface pressures north and south of the Equator in the western pacific. This may be due to Rossby waves generated



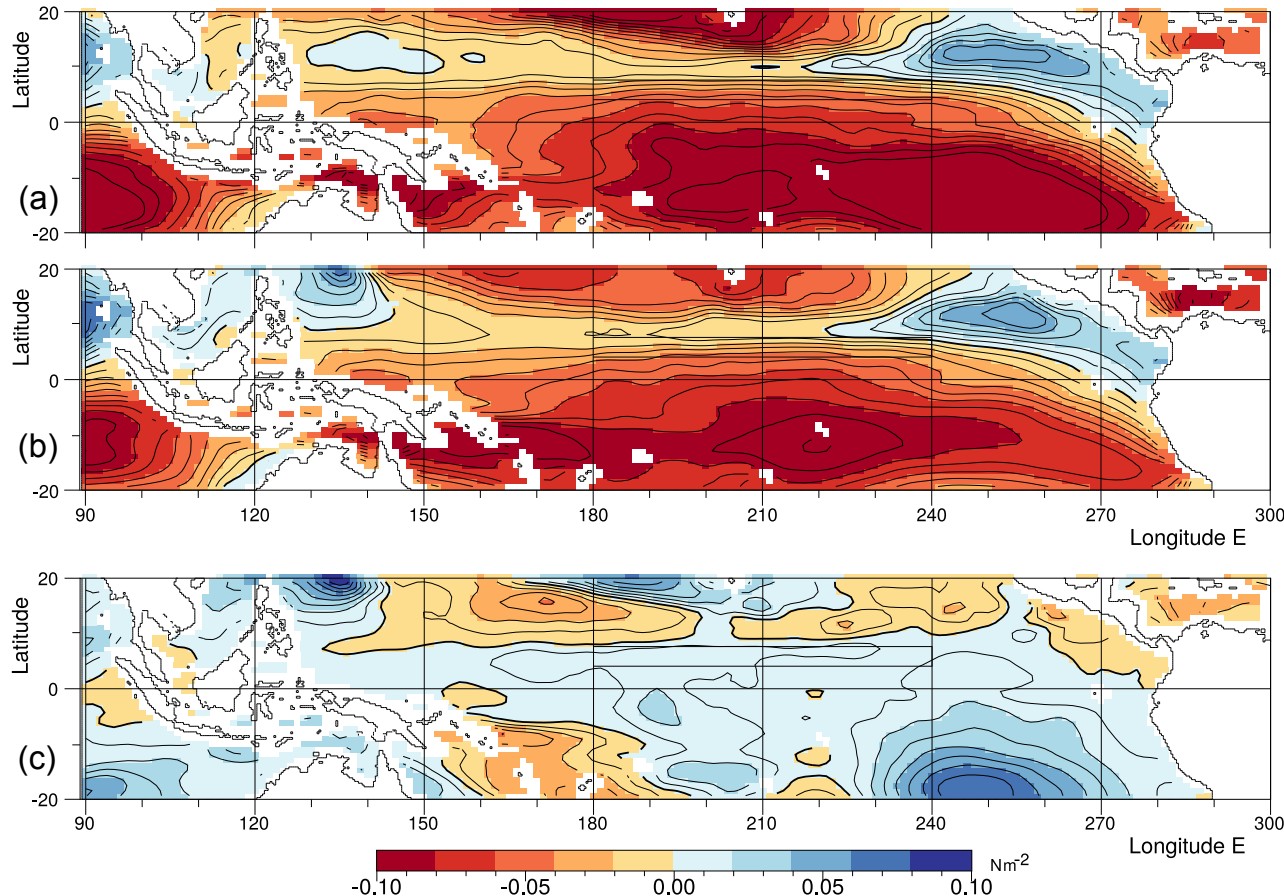

**Figure 4.** Averaged zonal wind stress acting on the ocean in September of year 5 for (a) the control run, (b) the forced run, both with contour interval $0.01\ \mathrm{Nm}^{-2}$ and (c) the difference, with interval $0.05\ \mathrm{Nm}^{-2}$.

by the increased ITCZ convection. The third route is via, what appears to be a change in the Hadley Circulation. This reduces the surface pressure in the Hadley Cell sinking region, and also results in reduce pressures along the Equator.

All three mechanisms affect pressure gradients along the Equator and so affect, via the zonal wind stress, the upwelling, structure and currents of the equatorial ocean.

### 3.4 Zonal Wind Stress acting on the Ocean

Figure 4 shows the zonal wind stress for the control and forced runs, in September of year 5, together with the difference between the two runs.



The stress acting on the ocean is primarily a function of the surface wind, positive and negative zonal stresses corresponding to westerly and easterly surface winds. The magnitude of the stress may depend to a certain extent on ocean currents, the surface wave field and near surface turbulence in the atmosphere, but usually such effects are small.

In both runs of the model, the equatorial Pacific is primarily a region of easterly winds forcing the ocean westwards. The easterlies are an extension of the south-east trades, part of the circulation around the high pressures of the South Pacific. In the north, the zonal wind stress is also westwards, due to the north-east trades associated with the high pressures of the North Pacific.

In between is the band associated with the ITCZ. In the central Pacific, this is a region of weak westward wind stress but
in the east and west, the wind direction changes. This is most noticeable off Central America where the zonal wind stress is strongly positive.

In the control run, the stress on the equator in mid-Pacific is typically -0.05 $\mathrm{Nm}^{-2}$. This a reasonably large value, the maximum zonal wind stress in the South Pacific between 30°S and the Equator being around -0.1 $\mathrm{Nm}^{-2}$.

In the forced run the zonal stress on the equator in mid-Pacific drops to around -0.04 $\mathrm{Nm}^{-2}$. The reduction is small but not
insignificant, given that this is a perturbation experiment.

The changes are clearer in Fig. 4c, showing the difference between the two runs. There is a broad band surrounding the Equator, where the response is positive, indicating weaker easterlies in the forced run.

Comparison with Fig. 2 indicate that the reduced wind stresses along the Equator are the result of three effects. In the east higher equatorial pressures are reduced because of the reduced strength of the sinking region in the South Pacific. This reduces
the strength of the south-east trade winds which are linked to the easterlies along the Equator.

In the central Pacific, the direct effect of increased convection along the ITCZ reduces pressure slightly, and so helps to reduce pressure gradient and the easterlies.

In the west the low initial pressures are raised slightly due to increased pressures in the western Pacific, north and south of the Equator. Again this may be a reflection of a changed Southern Oscillation. This pressure increase in the west, reduces the
pressure drop across the Pacific and so reduces the wind stress acting on the ocean.

To summarise this section, increasing the temperature of the NECC, has resulted in a significant reduction in the westerly wind stress acting on the equatorial Pacific Ocean. In the central Pacific this may be a result of deeper convection within the ITCZ. However near 190°E and 250°E, other process appear involved, the drop in surface stress around 250°E being connected with the changes further south in the Hadley Cell sinking region.

**4   The Response of the Ocean**

This focus on equatorial wind stress arises because it is the zonal wind stress at the Equator which is responsible for many of the ocean features associated with El Niños and La Ninas. This is partly because the equatorial scale of the atmosphere, discussed earlier, is larger than the 1 to 2 degree scale of the ocean.





On the Equator, the westward wind stress generates an Equatorial Current which carries warm surface water to the west.
This results in a deep surface thermocline in the west Pacific, the boundary with the cooler deeper waters sloping upwards from west to east.

The resulting slope in the density surface, generates a pressure gradient which drives an Equatorial Undercurrent. The current lies at a depth of only a few hundred metres and it brings cool water from deep layers in the west Pacific to much shallow depths in the east.

Within a degree on either side of the Equator, the Coriolis force starts being important and, where the wind stress is towards the west it results in an Ekman transport carrying the surface water away from the Equator. This is replaced by cool water from below, the final temperature of the water reaching the surface being a balance between Ekman driven upwelling and the downward diffusion of heat (Stommel, 1960; Webb, 2016). Together the processes are responsible for the cold pool of the eastern equatorial Pacific.

During El Niños, westward wind stress near the Equator is reduced. This results in a smaller thermocline slope, a reduced undercurrent, less upwelling, warmer water reaching the surface and a general warming of the cold pool.

In the west, the wind stress may change direction during El Niños. Ekman transport then converges warm surface water in the Equator, where the easterly surface current then carries it to a region of weak winds where deep atmospheric convection develops.

The previous section has shows that the equatorial wind stress is reduced during the forced run. The following two sections concentrate on whether this is sufficient to generate significant changes to both the ocean temperature structure and the zonal currents on the Equator. The final analysis section concentrates on the resulting changes in the ocean surface temperature field.

## 4.1  The Ocean Thermal Structure at the Equator

Figure 5, shows ocean potential temperature on the Equator in the control and forced runs, together with the difference. It
shows the pool of warm water in the western Pacific, and the shallow thermocline sloping up from depths of around 200 m in the west to depths close to the surface in the east.

The results are similar to the sections for August and October published by Johnson et al. (2002) and based on ten years of observations. At 160°E, both sets of figures show the thermocline at similar depths and have similar temperature profiles nearer the surface. Both also show the thermocline shallowing towards east with the 23°C contour nearing the ocean surface
around 240°E. However the mixed layer is much thicker in the model than in the observations.

Johnson et al. (2002) also show the change at each phase of the ENSO cycle. but without specifying the months involved. A comparison of the El Niño and la Nina phases show the surface thermocline shallowing, by ~10 m in the west, but becoming up to 40 m deeper in the central and eastern Pacific. In the western Pacific this results in a slight cooling of the surface layer. In the central and east surface temperatures increase by 2°C or more.

The present perturbation run is not expected to show large changes in thermocline depths and temperatures, but in the Fig 5 difference figure, surface temperatures are increased by up to one degree in the central and east Pacific. There is also a slight warming in the west.





**Figure 5.** Ocean temperatures in a section along the Equator averaged during September of year 5 for (a) the control run, (b) the forced run and (c) the temperature change due to the forcing.

At the level of the near-surface thermocline, there is a slight cooling in the west and a significant region of warming in the central and eastern Pacific. This indicates a shallowing of the thermocline in the west and a deepening in the central and eastern

Pacific.

## 4.2 Zonal Velocities at the Equator

Figure 6 shows ocean zonal velocity on the Equator in the control and forced runs, together with the difference. This time the agreement with the observations of Johnson et al. (2002) is not so good.



Both the control run of the model and the observations show a shallow surface Equatorial Current flowing westwards, and a deeper eastward flowing Equatorial Undercurrent. However the observations show the the surface current reversing west of the dateline in both August and October. This is not seen in the model.

At the depths of the undercurrent, the observations also show an undercurrent maximum near 155°E which is not seen in the control run. To the east, the observed velocities first reduce slightly before increasing, as in the model, to a maximum around 220°E.

In the Fig. 6 difference field, the response of the ocean appears to be split into two regions, with a boundary around 190°E. To the east, where the control run agrees best with the observations, there is a reduction in the strength of the Equatorial Current near the surface, and an undercurrent which moves deeper. The latter produces the band of negative velocity differences near 100 m and the wider band of increased velocities below.

In the west, the forced run results in a larger reduction in the speed of the westward flowing surface current. There is also a reduction in the speed of the eastward flowing undercurrent. Although the model does not appear to represent reality so well on the eastern side of the equatorial Pacific, both sets of changes are consistent with, but smaller than, the differences between the Johnson et al. (2002) El Niño and La Nina cases.

To summarise this and the previous section, there are noticeable difference between the control run of the model and the observed ocean, but the changes in the forced run are consistent with the changes that might be expected during a period of increasing El Niño index.

## 4.3 The Surface Temperature Field

Figure 7 shows sea surface temperatures in the control and forced runs, together with the difference. Surface temperatures in the band span the range 12°C to 30°C, a sharp contrast with the few degrees change in surface temperature that occurs during ENSO events.

Comparisons with observations (Levitus, 1982) indicate that the model is missing the east Pacific warm pool off Central America and replacing it with ocean temperatures that are too cold. Surprisingly Fig. 1 shows convection offshore from Central America in both runs. This is normally an indication of warm SSTs.

The cold ocean temperatures off Central America appear to be connected to both a North Equatorial Current, which is too cold, and a warm pool in the western Pacific, which too small and too cold. In marked contrast the temperatures in the South Pacific warm pool are consistent with observations.

In terms of the oceanography these deficiencies are serious. They almost certainly had some effect on the results presented in this paper.

The difference plot in Fig. 7, shows that during September, the forcing continues to warm the ocean over most of the forcing region, but that in the east, some cooling occurs. This results from the continued eastward advection of warm water in the control run that has not been allowed for in the forcing.

Elsewhere there is a cooling within a band just north of the forcing latitudes, and warming near the Equator, roughly coinciding with the forcing longitudes.





**Figure 6.** Ocean zonal velocities in a section along the Equator averaged during September of year 5 for (a) the control run, (b) the forced run and (c) the difference.

The reason for the cooling to the north is not known, but on the equator the temperature increase will be related to the surface warming discussed in the previous section. Here it is noticeable that the triangular shape of the region with increased temperatures is closely connected to the isotherms defining the equatorial cold pool. This again is similar to the warming associated with an El Noño like response.





**Figure 7.** Averaged sea surface temperature in September of year 5 for (a) the control run, (b) the forced run and (c) the difference.

# 5 Summary

## 5.1 Main Results

A forced increase in ocean temperature along the path of the North Equatorial Counter Current resulted in weaker easterly winds along the Equator, a change in the thermocline structure, a change in the undercurrent and increased sea surface temperatures in the region of the Equatorial cold pool.

All the changes were in the same direction as the changes associated with an increase in the Niño 3.4 index. The temperature changes were also of order 1°C, and thus compatible to the changes associated with the normal range of the Nino 3.4 index.





The increased ocean temperature did not result in any significant increase in low level convection along the line of the ITCZ.
However it did result in an increase in deep atmospheric convection. There was also a reduction in deep convection at other longitudes in the same latitude band, a band known to contribute to the rising branch of the Hadley Cell.

Atmospheric surface pressures dropped adjacent to the forcing region but this had little effect on pressure on the Equator. There was a larger reduction in surface pressures in the South Pacific, the maximum occurring in a region where the sinking branch of the Hadley Cell has a maximum.

The reduction in surface pressure reduced the strength of the south-east trades in the Pacific and thus the strength of the zonal wins along the Equator

## 5.2 Concerns

The model sea surface temperature field in the North Pacific is anomalous. The east Pacific warm pool off central America is missing and the North Equatorial Current is too cold. The west Pacific warm pool is cool and of restricted size.

This is unfortunate and is a problem that needs to be addressed in future tests. However there is no evidence that the error has affected the main results of this paper which concern changes occurring nearer the Equator and south of the Equator.

The results show an El Niño type response in the central Pacific, but it is insufficient to generate westerly winds in the western Pacific and there is no evidence of a deep convection developing on the Equator near 180°E, both features associated with El Niños.

This may be because the magnitude and region of forcing was too restricted or because sea surface temperatures in the west Pacific were too low. It is an aspect needing further study.

## 5.3 Mechanisms

It was initially expected that increased convection along the line of the ITCZ would reduce sea surface pressure and, through an increased equatorial Rossby radius, affect surface pressures and winds on the Equator. The results showed that low level
convection did not increase. The was an increase in convection higher in the atmosphere up to and passing 300 hPa.

One result of this was a small reduction in sea level pressure along the line of the ITCZ, but this reduction had little effect on sea level pressures at the Equator.

Instead the most important effect of increased deep atmospheric convection in the ITCZ appears to have been in changing the longitudinal structure of the Hadley Cell. As reported by Webb (2023), the structure of the rising branch of the Hadley Cell
may involve convection to mid-atmospheric levels by the ITCZ, this air being later entrained by plumes at other longitudes to levels above 300 hPa.

The increased deep atmospheric convection by the ITCZ, will have affected the pattern of divergence above 300 hPa and appears to have affected the patten of Hadley Cell sinking. The model results indicate that sinking is reduced in the South Pacific. The resulting reduction in surface pressure, reduces the strength of the south-east trades, reduces the easterlies on the
equator and results in the El Niño response of the equatorial ocean.



The pressure change generated by the forced run also have similarities to the structure of the Southern Oscillation which is statistically linked to the oceanic El Niños.

## 5.4 Hypothesis

The original hypothesis was "that the changes observed during El Niños are, at least in part, a response of the coupled ocean/atmosphere system to changes in sea surface temperature along the path of the North Equatorial Counter Current". It is possible that the term "El Niños" should be replaced by "ENSO" but whatever is chosen, in physical systems such statements can never be proved correct. In contrast, to be proved wrong just one counter-example is sufficient.

So to conclude, in an initial test, using a respected climate model, the hypothesis has not been proved to be wrong.

*Acknowledgements.* The present study was carried out at the National Oceanography Center, UK, and funded by the Natural Environment Research Council grant NE/Y005589/1. I would like to acknowledge the generous support of NOC and its staff, especially the members of the MSM group.

I would also like to acknowledge previous support, especially that concerned with my interest in the equatorial Pacific, by the UK Institute of Oceanographic Sciences and the CSIRO Division of Fisheries and Oceanography.

The study made use of version 2.1.3 of the CESM2 climate model. The model is made available and supported by the U.S. National Center for Atmospheric Research under the sponsorship of the National Science Foundation. Thanks to all the scientists, software engineers and administrators who contributed to the development of CESM2.

*Code availability.* The CESM model is available from NCAR. A copy of the modified ocean model subroutine used for the forced run is available from "https://github.com/djwebb/ccode".

*Author contributions.* N/A - single author paper

*Competing interests.* The author, with Prof. J. Johnson, was a founding editor of Ocean Science.



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
