# Peer review of "On the Response of the Equatorial Atmosphere and Ocean to Changes in Sea Surface Temperature along the Path of the North Equatorial Counter Current"

_EGUsphere, 2024_

## Referee Comment (RC2)

**Review of "On the Response of the Equatorial Atmosphere and Ocean to Changes in Sea Surface Temperature along the Path of the North Equatorial Counter Current" by David J Webb**

**Paper Summary**

This paper continues work in previous papers investigating the hypothesis that the North Equatorial Counter Current (NECC) plays an important role in the development of El Nino events. The CESM climate model is used to assess the impact of a warm anomaly in the sea surface temperature in the NECC region, that is imposed from August onward, on the atmosphere and ocean during September, 4-8 weeks after the anomaly is first imposed. The warm anomaly strengthens deep atmospheric convection locally and appears to affect the strength and longitudinal structure of the Hadley cell. It appears to have a large impact on the atmospheric pressure fields across the entire Pacific ocean (to high latitudes) and weakens the surface pressure gradient along the equator resulting in weaker easterly winds near the equator, reduced Ekman pumping away from the equator and SSTs that are about 1°C warmer in the central Pacific than in the control.

**Overall comments**

My knowledge of the literature in this field is by no means comprehensive and perturbation experiments of this sort are not unusual. Notoriously, the ITCZ in the southern hemisphere is too strong in nearly all climate models (an issue known as the double-ITCZ problem). So there may be a literature of papers describing similar experiments, but I am not aware of it and I suspect that the author would have searched the literature quite diligently for such paper. So I think that this could potentially be quite an unusual and valuable paper.

The comments below are divided into 3 sections:

- questions about the scientific interpretation
- suggestions on where the presentation might need to be improved or how it might be improved
- bullet points on typos – these are very brief.

I've made a lot of comments so I have said that "major revisions" are required, but I believe that most of the points will not be difficult to respond to.

**Comments on the science**

L 29; "easterlies on" should be "easterlies near" because the Ekman transports are off the equator.

L 42: The double ITCZ problem is notorious so surely it has been intensively studied (this problem is mentioned later in the paper). A brief review of papers in that literature that relate to this paper might be appropriate.

L78-79: "will lower". Is this obvious / well known ? Could you provide a reference to a paper or textbook on this point?

L 119: Graham (2014) could be cited here.

Graham, T., 2014: The importance of eddy permitting model resolution for simulation of the heat budget of tropical instability waves. *Ocean Modelling,* **79**, 21‑32, https://doi.org/10.1016/j.ocemod.2014.04.005.

L 129: Surely some other authors could be cited here;  The phase locking of El Nino is well known. To my mind, the maximum growth rate being in September gives some credibility to the hypothesised NECC mechanism as the ITCZ is presumably strongest at that time of year.

L 142-145: The forced runs only involve 2 months of integration. It would be good to check the robustness of the results  by performing 2 months of integrations starting in August of other years (e.g. years 4, 5, 6, 7, 8, 9, 10). Some of these years might be El Ninos or La Ninas so some other design with perhaps a 20 year control simulation to branch from might be required.

L 154-156: It looks to me that the field increases significantly just north of the forcing region from about 205 to 240 E. The description given doesn't really say that?

 L 162: Looking at Fig 1, 20N is actually quite a long way north. I would say there is more than one thing going on within that range.

Table 1: The zonal average values are remarkably independent of height. If this is the model's vertical velocity variable it does not look like the vertical velocity for mode 1 which has zero vertical velocity at the tropopause and sea surface.

L 189-191: The mid-troposphere maximum vertical velocity projecting strongly onto the first mode is what one would expect; so that is reassuring.

L 203-204: The depth of the valley looks nearer 0.25 C on the eastern side of the forcing region.

Figure 3, L 223, L 229: The very large scale and amplitude of the surface pressure response is interesting. I would say the pressure changes are larger than 10 hPa over quite large areas. The response within 20 degrees of the equator is much weaker than that further poleward. Large teleconnections between Nino3 SST and pressure, SST at high latitudes are well known but I wonder if this aspect of the structure has been seen before. If this is related to the latitude of the descending branch of the Hadley cell I had not realised that that branch is so far poleward. Perhaps one could see that in a poleward extended version of fig 1?

L 241 "show a maximum". Isn't it a minimum in Fig 3?

L 254: the surface wind stress is a turbulent stress so the near surface stratification and turbulence affects the exchange coefficient.

L 287: I'm not sure the EUC dynamics is quite that simple. Vallis (2017) sec 22.3 describes a more complex dynamics.

L 295-296: on the timescales studied in this paper (1-2 months) some of these mechanisms will barely get started. Rather than "warmer water" I would be inclined to say "less cold water" (a smaller flux of cold water, or water that is slightly less cold).

L 307: Johnson et al 2002 is rather old now. There are 20 more years of TOGA TAO data available. Is this still the best source of cross-sections published?

L 325: This is a good/interesting point! I was not previously aware of it.

L 339: comparing the current changes with those in the Johnson paper, I'm not very convinced that the changes in the forced run are that consistent with El Nino changes. Could you re-visit that please?

L 348-349: This sentence seems a bit speculative. Is it really needed here?

Figure 6c: The contour interval doesn't look right

Lines 358-359: I wonder whether the cooling to the north can be related to Ekman pumping; one could check whether the spatial extent and sign are consistent with that;  there is rather substantial warming on the equator and the warming is a maximum just north and south of the equator – isn't that worth saying?; "the previous section" doesn't seem to refer back correctly.

L 364-366: A brief summary of the experiment and the short period of the forced integration would be appropriate. On the equator the wind and SST changes are the most important; the slight deepening of the thermocline is also worth mentioning.

L 374: please confirm that the sinking branch of the Hadley cell is this far south (see the earlier point)

**Suggestions on presentation**

Abstract: Line 7: "Both mechanisms" – The first mechanism seems to be the local effect on the surface pressure in the ITCZ. But that does not change the surface pressure gradient along the equator. So some re-writing is needed?

General point: Many of the paragraphs are rather short and make a single point. I quite often felt that the train of thought would be easier to follow if two or three paragraphs were joined together. I've noted some examples of this below.

Lines 18-22: The ascending and descending branches of the Hadley cell must be quite closely related. So I feel somewhat sceptical about the point made in lines 20-22. This is only the introduction so perhaps I am being a bit over critical.

Line 29: No new paragraph here? See general point

L 38: No new paragraph here

L 44: No new paragraph here?

L 47: Actually the setting/context for the present paper

L 51: No new paragraph here? Use "It" instead of "The study"

L 55: replace "extracted" by "exported"?

L 70-73: These are very important points. Together with panels a and b of Figure 7, they seem to me to be quite key to the paper. The SST relative to its neighbourhood seems to be what is important for triggering deep convection. I think the paper might come across better if Figure 7 was brought forward to Figure 1.

L 83: It would be really helpful to refer to a specific sub-section of Vallis. The paper does not have too many figures and it might add considerably to the paper if a figure showing a map of the idealised response were included.

L 89: No new para here

L 91-92: It might be worth emphasising that the equator is rather cold at these longitudes (Fig 7a, 7b) so similar temperature perturbations on the equator will not trigger similar convective motions.

L 132-135 and 139-140. This description of the forcing feels very ad hoc and I don't think one could reproduce the forcing from its description. The sentence in L 139-140 should appear immediately after L 135. I think the reader does not know at this stage that the forced integrations only span August and September. This makes the description seem very obscure.

L 136. The decay time-scale of 2 days is very short. Is it possible to say how much heat is being input into the ocean, perhaps as monthly averages of Watts per square metre?

L 142-145: This is a really important paragraph. The length of the forced integration (2 months) should be highlighted in section 2.1 and in the summary section. This is sufficient time for the atmospheric response but only for the initial ocean response.

L 147 and Figure 1. Is this quantity the "vertical" velocity (omega) of the atmospheric model? It would help ocean readers to be spoon fed on this point.

L 151: It is asking a lot of the reader to follow up the NOC report. Could the ERA5 results not be included in the figure? If that is not worth doing, is it worth making the point at all?

Table 1 and its discussion: The zonal average values seem to me to be just a distraction. They are hard to understand and not very relevant to the discussion. Why not just omit them? The 180-240 values are what matters.

L 205-207. I find this para hard to follow.

L 212-14: This was hard to follow on first reading. I think seeing Fig 3 and reading section 3.3. helps understand it. If the response to an ITCZ-like line of convection had been presented earlier it might be easier to follow the argument; I don't and many others won't remember Gill's solution for off-equatorial forcing, particularly the signs of the response. It would help to say we are considering the response to the west of the forcing region. That is implied by talking about the Rossby wave response but saying it explicitly would help many readers.

L215: No new para. We need to look at Fig 3 to see the maximum is at 25S.

L 276: "westerly" isn't it an easterly wind stress (from the east)

L 278: Why 190E and 250E in particular? I think there might be a typo here.

L 327: I consulted Johnson et al 2002 because I was confused by the intended meaning of this point on first reading. This maximum is just a local maximum; I'm not sure that the model missing it is particularly concerning. It is a minor point that makes the paper a bit harder to read.

L 345: It would be easier for the reader to follow this point if an SST reanalysis were included in the figure. There are better SST analyses readily available than Levitus 1982.

L 353-355: Why "continues" that seems a strange word to use and hence disconcerting for the reader. "some cooling" – there seems to be a small region of very intense cooling? "that has not been allowed for" – I'm not sure what that is intended to mean, it probably needs rewriting

L 368: "compatible" does not seem the right word; "comparable" or "of similar magnitude"?

L 380: "future tests" Do you mean using other CMIP models with different biases?

L 389: "an increased Rossby radius"; I don't follow what this is intended to mean

L 408: Karl Popper would be delighted to see such a statement. But most scientists would not see it as very helpful. It might be worth re-considering this point.

**Bullet points on typos**

L 14: insert work after early

L 33: interms -> in terms

L 62: delete "of" before "across"

L 112: should be "one of the"

L 114: settled

L 122: coupled

L 195: delete "lea"?

L 202: "a narrow"

L 208: comes

L 210: "much"

L 300: "shown"

L 376 "winds"

L 390 "There"

L 398: "pattern"

M. J. Bell